# Detubularized Ureterosigmoidostomy for the Creation of Continent Neobladder in Children: Cases Report and Review of the Literature

**DOI:** 10.3390/children8040279

**Published:** 2021-04-05

**Authors:** Edoardo Bindi, Michele Ilari, Giovanni Torino, Francesca Mariscoli, Fabiano Nino, Giovanni Cobellis, Ascanio Martino

**Affiliations:** 1Pediatric Surgery Unit, Salesi Children’s Hospital, 60123 Ancona, Italy; michele.ilari@ospedaliriuniti.marche.it (M.I.); giovannitorino1@libero.it (G.T.); francesca.mariscoli@ospedaliriuniti.marche.it (F.M.); fabiano.nino@ospedaliriuniti.marche.it (F.N.); giovanni.cobellis@ospedaliriuniti.marche.it (G.C.); ascanio.martino@ospedaliriuniti.marche.it (A.M.); 2Faculty of Medicine and Surgery, University Politecnica of Marche, Via Tronto, 10/a, 60126 Ancona, Italy

**Keywords:** continent urinary diversions, ureterosigmoidostomy, cystectomy, Mainz pouch II, pediatric surgery

## Abstract

**Introduction**: To report our experience in continent urinary diversions, we describe two cases we treated performing detubularized ureterosigmoidostomy. In children, in the case of malformations or neoplastic diseases affecting the bladder, the need for a cystectomy is not so frequent. When cystectomy becomes mandatory, there is a need to create a continent bladder diversion. Mainz pouch II and Cologne pouch are procedures that utilize a detubularized sigma as a reservoir in order to build up a continent neo-bladder. **Materials and methods**: This is a retrospective study performed at the Pediatric Surgical Unit of the Salesi Children’s Hospital. In this work, we reviewed data about two patients who underwent surgery for the creation of a sigmoid neo-bladder by the Mainz pouch II and Cologne pouch techniques. **Results**: In our experience, we treated a girl who was affected by a bladder’s rabdomiosarcoma and a girl born with a bladder exstrophy and treated at birth abroad. In both patients, a complete cystectomy was performed and a continent neo-bladder was created by a detubularized ureterosigmoidostomy. In the first case, we performed the Mainz pouch II technique and in the second, the Cologne pouch technique. **Discussion:** Different techniques have been developed with the main goal of the creation of an orthotopic neo-bladder, which has to be a low pressure reservoir with a continent sphincteric mechanism. Detubularized ureterosigmoidostomy is a good choice in pediatric patients. Our study, according to other works, shows that these procedure are safe with good long-term outcomes.

## 1. Introduction

In pediatric surgery, in the case of malformations or neoplastic diseases affecting the bladder, the need for a cystectomy is less and less frequent. In cases where cystectomy becomes mandatory, there is a need to create a continent bladder diversion. Mainz pouch II and Cologne pouch are procedures that utilize a detubularized sigma as a reservoir in order to build up a continent neo-bladder. In this work, we report our experience in two patients treated with these techniques.

Urinary bladder substitution is not a common procedure in children, for whom every effort is made to preserve the native bladder. However in the case of malignancies, such as rabdomiosarcoma, or malformations, such as a bladder’s exstrophy, it is not always possible to preserve the native bladder.

The improvement of medical therapies and surgical procedures has reduced significantly the rate of cystectomy for bladder tumors in children. Nevertheless, in some patients, after chemotherapy and radiotherapy the bladder becomes fibrotic and functionally compromised, with no possibility to save it [1]. Moreover cystectomy can become mandatory in a bladder’s exstrophy when the bladder plate is small. presenting polypoid changes of the mucosa, with the risk of malignant transformation [2,3].

Several techniques allow the creation of a competent neo-bladder. Mainz pouch II and Cologne pouch are procedures that utilize a detubularized sigma as a reservoir. Several series reported good outcomes using these procedures. In this work we report our experience in two patients treated with these techniques.

## 2. Materials and Methods

This is a retrospective study performed at the Pediatric Surgical Unit of the Salesi Children’s Hospital. In this work, we review data about two patients who underwent surgery for the creation of a sigmoid neo-bladder by Mainz pouch II and Cologne pouch techniques.

A retrospective clinical-records review was performed on two patients operated on for the creation of a sigmoid neo-bladder by the Mainz pouch II and Cologne pouch techniques at the Pediatric Surgical Unit of the Salesi Children’s Hospital in Ancona. We collected data about the demographic characteristics, bladder pathology, diagnostic examinations, operative procedures, post-operative complications, electrolytic balance and urinary continence outcomes.

## 3. Results

In our experience, we treated a girl who was affected by a bladder’s rabdomiosarcoma and a girl born with a bladder exstrophy and treated at birth abroad. In both patients, a complete cystectomy was performed and a continent neo-bladder was created by a detubularized ureterosigmoidostomy. In the first case, we performed the Mainz pouch II technique and in the second, the Cologne pouch technique.

### 3.1. Case 1

A two-year-old girl came to our unit with abdominal pain and hematuria. A cystoscopy was performed, with evidence of a mass originating from the vesical trigone. We collected biopsies of the mass. The hystological diagnosis was a rabdomiosarcoma of the bladder. The patient started a neoadjuvant therapy and then, according to protocol, she underwent surgery. A cystectomy with the creation of a continent neo-bladder by the Mainz pouch II technique was performed (Figure 1A,B).

The post-operative course was free from complications and the patient was discharged, entering our follow-up protocol (Figure 2A,B).

Every two years, the patient underwent endoscopy with random biopsies to evaluate the condition of the ureterosigmoidostomy.

After two years, the patient was in good condition with no complications about the pouch.

During the follow-up, the patient developed a right hydroureteronephrosis due to a stenosis of the right ureterosigmoidostomy with a decrease of the right renal function. A redo ureterosigmoidostomy was performed with the complete resolution of the stenosis.

To date at the last check, at 13 years of age, the patient presented in good condition, with good continence, no electrolyte imbalances, no urinary infections and blood pressure to normal values. The patient also had a MAG3 renal scan annually, which showed good renal function and no scarring.

### 3.2. Case 2

An 11-year-old girl came to our attention for a complete urinary incontinence. The patient, born abroad, presented at birth a bladder’s exstrophy that was corrected soon after with the closure of the bladder’s plate.

A functional examination and a renal scintigraphy were performed, with evidence of a bilateral third degree vescicoureteral reflux with good renal function.

The patient underwent a cystectomy and a Cologne pouch intervention with a complete cystectomy and creation of a continent neo-bladder with the sigma (Figure 3, Figure 4, Figure 5, Figure 6 and Figure 7).

At 12 years of age, during follow-up, the patient did not present long-term complications. Urinary continence was good, with a neo-bladder emptying nearly every 2–3 h (Figure 8A,B). The patient is clean during the day and she presents a good quality of life.

There were no post-operative complications and the patient was discharged in good conditions.

## 4. Discussion

In pediatric surgery, bladder pathologies that can lead to a severe impairment of its function and to urinary incontinence are rare. Every effort is made to save the native bladder, but in some cases a cystectomy is mandatory, with the consequent need to build up a continent neo-bladder. Different techniques have been developed, with the main goal the creation of an orthotopic neo-bladder, which has to be a low pressure reservoir with a continent sphincteric mechanism. Detubularized ureterosigmoidostomy is a good choice for the creation of a continent neo-bladder in pediatric patients. Our study, according to other works, shows that these procedure are safe with good long-term outcomes.

In pediatric surgery, bladder pathologies that can lead to a severe impairment of its function and to urinary incontinence are rare but challenging. In these cases, every effort is made to save the native bladder. In some cases a cystectomy is mandatory, with the consequent need to build up a continent neo-bladder. The main indications for this kind of surgery are a bladder’s neoplasms and major malformations, such as the bladder’s exstrophy and exstrophy of cloaca. In the first case, a cystectomy is indicated when the pathology is advanced, or when, following radiation therapy, the bladder has become fibrotic, losing its continence.

In the case of major malformations such as bladder’s exstrophy with polipoid changes, it is sometimes not possible to reconstruct a functioning bladder, and urinary diversion remains the only possibility.

In 1852, Simon first described the diversion of urine into the rectum [4,5] but problems soon arose associated with this procedure. In fact, ascending urinary infections with renal injury, metabolic imbalances such as hyperchloraemic acidosis and long-term complications such as anastomotic neoplasms led to the abandonment of ureterosigmoidostomy in favor of ileal conduit urinary diversion in the mid-20th century [6,7]. Nevertheless, over the years, an improved understanding of the pathophysiology of hyperchloraemic metabolic acidosis, the development of alkalinizing drugs and the antirefluxive technique of ureteric implantation solved many of the previous problems of ureterosigmoidostomy and reawakened interest in this procedure [8].

Different techniques have been developed over time, but the main goal of these procedures is the creation of an orthotopic neo-bladder, which has to be a low pressure reservoir with a continent sphincteric mechanism [9,10].

Bladder substitution in children may be performed by using either the small bowel or the colon. Both choices have as their main point the principle of detubularization. In fact, the conversion of a tubular segment of intestine into a spherical reservoir of bowel decreases the pressure and peristalsis and increases the capacity of the segment. A detubularized segment presenting a more spherical configuration results in a lower pressure reservoir with a minor absorption area, thus minimizing the appearance of ureteral reflux and metabolic complications. The other point is the importance of preventing the reflux of urine. The presence of urine reflux leads to ascending infections with serious renal injuries and a decrease in its function. The principles of tunnel or nipple valves are used for reflux prevention in uretheral intestinal implantation and are similar to those used for vesicoureteric reimplantation. Structural differences in the bowel wall may involve some procedural variations; however, the principles of the procedures are technically similar in that filling of the reservoir compresses the ureteral lumen, thus preventing the reflux of urine [11].

These principles were applied to the ureterosigmoidostomy technique to create a rectosigmoid pouch (Mainz pouch II) in 1990 [12]. In the literature, there are several series that reported good outcomes with these procedures.

Mingin et al. [13] described five patients with bladder exstrophy who underwent the Mainz pouch II procedure between 1996 and 1998 to create a rectosigmoid pouch, allowing urine to drain into and be eliminated via the rectum. Three patients required oral sodium bicarbonate to correct a metabolic acidosis, but upper tracts remained non-dilated; all patients were continent during daytime and nighttime, and there were no episodes of pyelonephritis during follow-up, which was between about one and three years.

In addition, Rubenwolf et al. [14] recently reported on a long-term follow-up after continent anal diversion, and described a cohort of 82 exstrophy patients who underwent continent anal urinary diversion between 1970 and 2015 (57 were eligible and 32 of them chose to participate). The median age was 38.6 years. The group reported that 20% of upper tract complications occurred after a Mainz pouch II procedure, and the other 80% of complications occurred after traditional ureterosigmoidostomy.

In a similar way, there are studies about patients treated for bladder exstrophy in which the creation of a continent urinary diversion by the Cologne pouch technique has been demonstrated to have good results. Klein et al. treated 29 patients affected by bladder exstrophy with a continence rate of 81.5%, and a low rate of complications concerning the urinary tract [15].

The results presented in our work, although limited to a selected few cases, are comparable with those of other studies in the literature. The outcome concerning urinary continence shows that these patients are clean during the day, with an acceptable time between each bladder emptying, resulting in a good quality of life, which is essential considering the age.

In our series there were no cases of metabolic imbalances, due to the fact that in our follow-up protocol, every patient was treated with carbonates administration.

As a main complication, we report a right severe hydronephrosis with a decrease in renal function caused by a stenosis in the ureteric anastomosis. Unfortunately, these kinds of complications are not predictable, but they are not related to the technique itself. Conversely, we did not find any cases of urinary infections, confirming the good outcome of this procedure in terms of antirefluxive results.

In this kind of surgery there is the concern of a higher risk of later malignancy, secondary to the mixture of urine and feces, which must be taken into account. In Rubenwolf et al. [14], 8 out of 82 patients (9.8%) developed cancer following continent anal diversion, and the diagnosis of the cancer ranged from 15 to 48 years at a median of 34 years after creation of the diversion [16]. Currently, two open randomized trials, one in the UK (ClinicalTrials.gov Identifier: NCT03049410) and another in Italy (ClinicalTrials.gov Identifier: NCT03434132) plan to compare outcomes for bladder cancer patients undergoing a radical cystectomy with either intracorporeal or open urinary diversion [17].

The endoscopic examination of our patients has not found to date any pathologic changes, but a strict follow-up is mandatory in order to avoid further severe problems. Our protocol, which provides for an endoscopic examination with biopsies every two years, has proven to be effective in identifying any post-operative complications. Obviously, in the case of the aforementioned symptoms for ureteral reflux or for possible transformation of the periureteral tissue, these must lead to a prompt colonoscopic examination.

In recent years, pediatric surgery has seen great development in minimally invasive surgery that also has a good application in the urological field [18]. The laparoscopic and minivasive approach could also lead to good results in this type of surgery, as has been demonstrated in other pathologies [16,19]. However, the main indication currently remains that of an open approach that guarantees a good view of the structures and, especially in the case of tumor pathology, the reduction of the risk of spilling.

## 5. Conclusions

Detubularized ureterosigmoidostomy is a good choice for the creation of a continent neo-bladder in pediatric patients. Our study, according to other works, shows that these procedure are safe with good long-term outcomes.

In addition, because these techniques require no devices (bags or catheters) they are particularly suitable for patients from poor countries.

Nevertheless, we include the Mainz II and Cologne pouch among the choices offered to patients requiring cystectomy and continent urinary diversion.

## Figures and Tables

**Figure 1 children-08-00279-f001:**
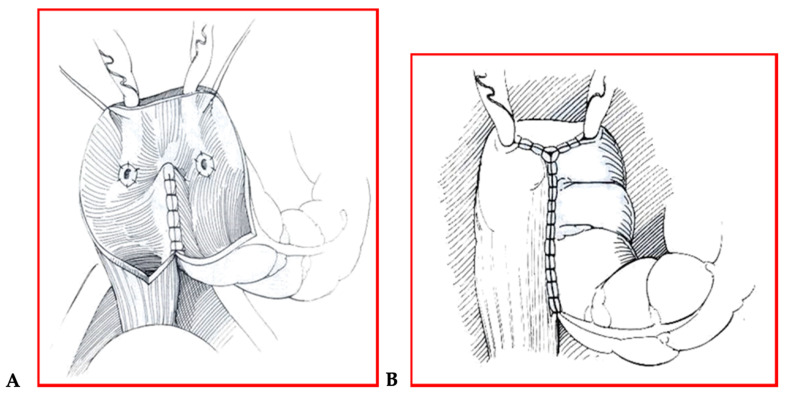
(**A**,**B**) Mainz pouch II technique.

**Figure 2 children-08-00279-f002:**
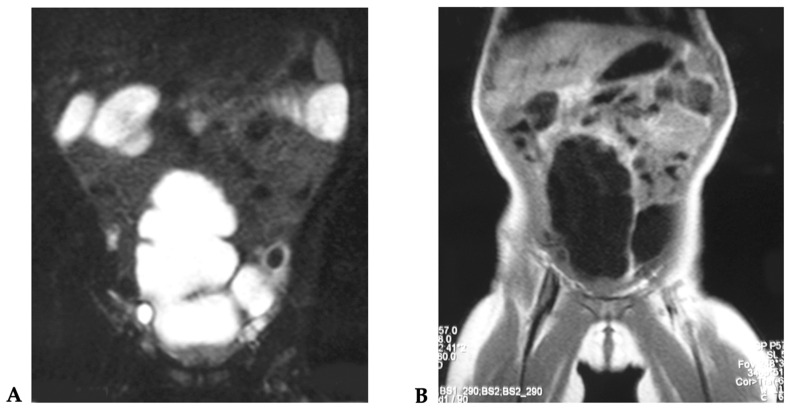
(**A**,**B**) MRI examination after surgical intervention.

**Figure 3 children-08-00279-f003:**
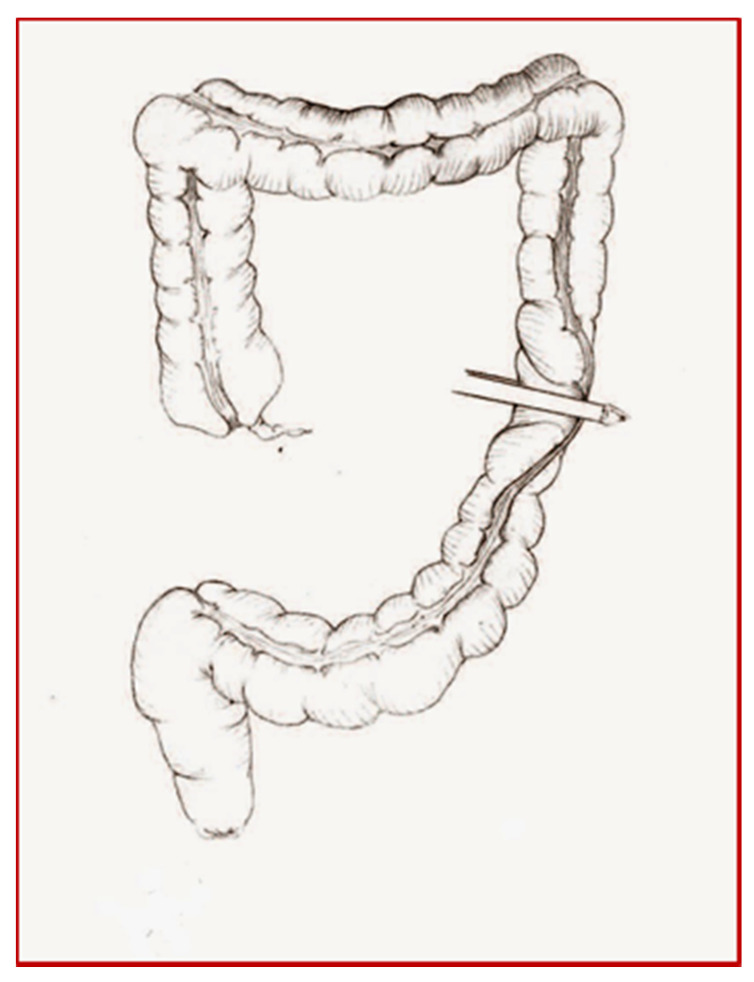
Cologne pouch technique: section of the left colon.

**Figure 4 children-08-00279-f004:**
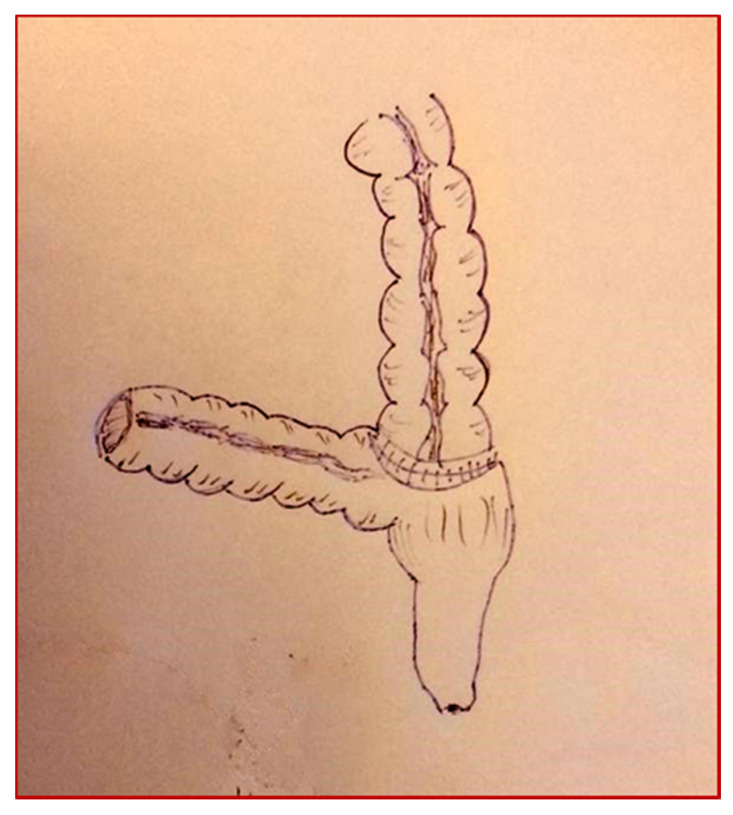
Cologne pouch technique: end-to-side colo-rectal anastomosis.

**Figure 5 children-08-00279-f005:**
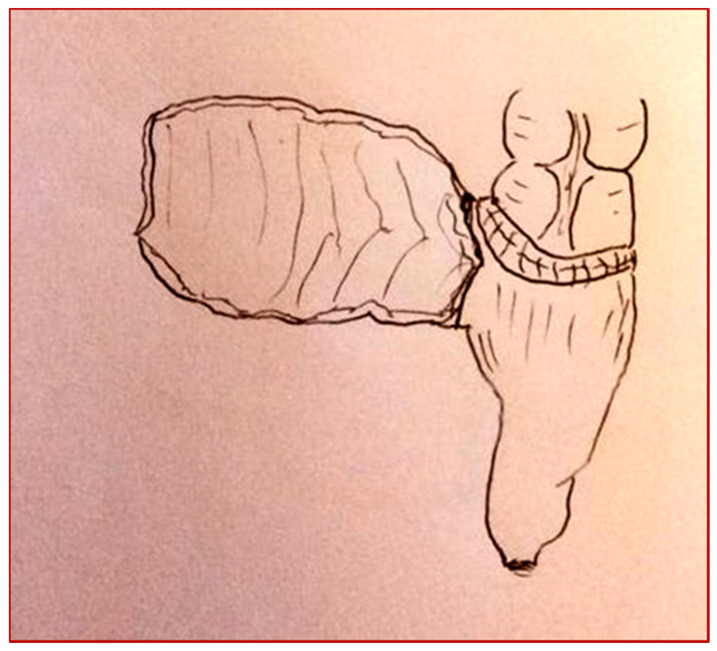
Cologne pouch technique: intestinal detubularization and creation of the pouch.

**Figure 6 children-08-00279-f006:**
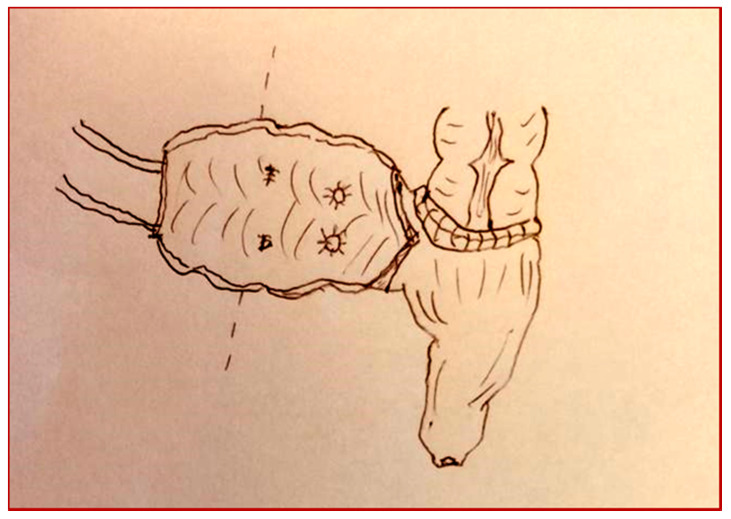
Cologne pouch technique: reimplantation of the ureters with an antirefluxive technique.

**Figure 7 children-08-00279-f007:**
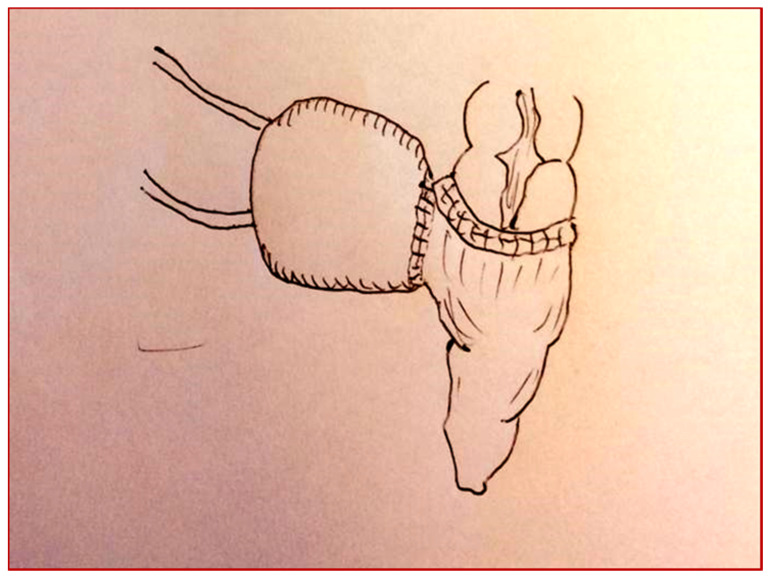
Cologne pouch technique: creation of the pouch.

**Figure 8 children-08-00279-f008:**
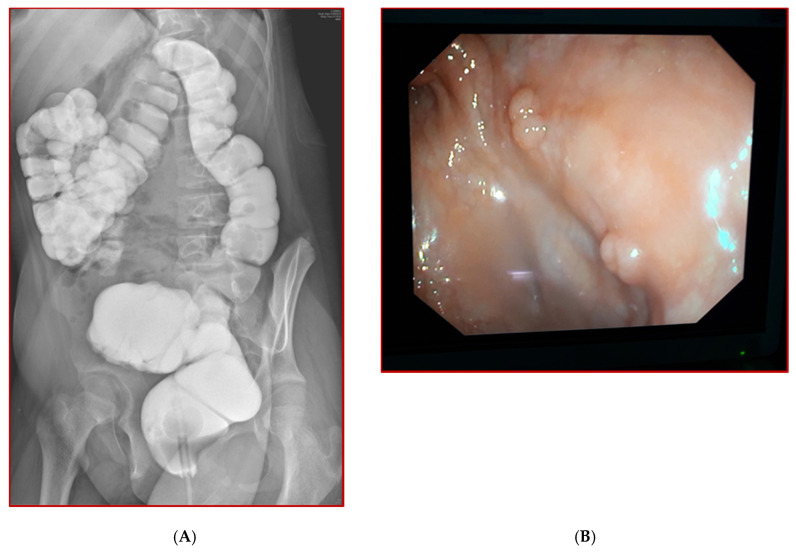
Barium enema (**A**) and endoscopy (**B**) examinations during follow after Cologne pouch intervention.

## Data Availability

Not applicable.

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
