# Peer review of "Detubularized Ureterosigmoidostomy for the Creation of Continent Neobladder in Children: Cases Report and Review of the Literature"

_children, 2021, doi:10.3390/children8040279_

Round 1

Reviewer 1 Report

The authors present two case reports of children undergoing cystectomy and colonic urinary diversion.  They then report a brief history and literature review of colonic urinary diversion in children.  The length of this review appears appropriate to go with a case report.  There are some english grammar errors which could be corrected.

Author Response

Thanks a lot for your suggestions. The manuscript has been entirely revised by an English native speaker colleague and it has been corrected. 

Regards,

Dott. Edoardo Bindi

Reviewer 2 Report

This description of two cases may be of help for some surgeons. A few comments: some elements are included in the discussion that should actually be included in the results (per individual): e.g., continence, micturition frequency (and manner) including volumes (and defecation, where relevant), incidence of urinary tract infections, more insight into the results of kidney function (and or dilatation or scarring) and metabolic function (and thus also relevant medical management per individual) and also (child to adult) physical development. I also think it would be good to mention what exactly you biopsied each time; normal looking tissue? random biopsies? or specific? and the rationale of the chosen strategy.

Author Response

Thanks a lot for your careful evaluation. I appreciated your considerations and suggestions. In the discussion we described the results already reported in the “Results” section. Anyway we clarified in a better way these data, adding an explanation in the “Results” section. We also added informations about the biopsies we took in examination.

Regards,

Dott. Edoardo Bindi